# A novel cell culture system modeling the SARS-CoV-2 life cycle

Xiaohui Ju[1], Yunkai Zhu[2], Yuyan Wang[2], Jingrui Li[3], Jiaxing Zhang[4], Mingli Gong[1], Wenlin Ren[1], Sai Li[4,5], Jin Zhong[6], Linqi Zhang[1], Qiangfeng Cliff Zhang[4,5], Rong Zhang[2], Qiang Ding[1,5]*

**1** School of Medicine, Tsinghua University, Beijing, China, **2** Key Laboratory of Medical Molecular Virology (MOE/NHC/CAMS), School of Basic Medical Sciences, Shanghai Medical College, Biosafety Level 3 Laboratory, Fudan University, Shanghai, China, **3** State Key Laboratory of Plant Physiology and Biochemistry, College of Biological Sciences, China Agricultural University, Beijing, China, **4** School of Life Sciences, Tsinghua University, Beijing, China, **5** Beijing Advanced Innovation Center for Structural Biology, Tsinghua University, Beijing, China, **6** Unit of Viral Hepatitis, CAS Key Laboratory of Molecular Virology and Immunology, Institut Pasteur of Shanghai, Chinese Academy of Sciences, Shanghai, China

* qding@tsinghua.edu.cn

**Data Availability Statement:** RNA-seq dataset generated here can be found in the aforementioned NCBI Gene Expression Omnibus (GEO Accession no. GSE162629, https://www.ncbi.nlm.nih.gov/geo/query/acc.cgi?acc=GSE162629).

## Abstract

Severe acute respiratory syndrome coronavirus 2 (SARS-CoV-2) causes the global pandemic of COVID-19. SARS-CoV-2 is classified as a biosafety level-3 (BSL-3) agent, impeding the basic research into its biology and the development of effective antivirals. Here, we developed a biosafety level-2 (BSL-2) cell culture system for production of transcription and replication-competent SARS-CoV-2 virus-like-particles (trVLP). This trVLP expresses a reporter gene (GFP) replacing viral nucleocapsid gene (N), which is required for viral genome packaging and virion assembly (SARS-CoV-2 GFP/ΔN trVLP). The complete viral life cycle can be achieved and exclusively confined in the cells ectopically expressing SARS-CoV or SARS-CoV-2 N proteins, but not MERS-CoV N. Genetic recombination of N supplied *in trans* into viral genome was not detected, as evidenced by sequence analysis after one-month serial passages in the N-expressing cells. Moreover, intein-mediated protein trans-splicing approach was utilized to split the viral N gene into two independent vectors, and the ligated viral N protein could function *in trans* to recapitulate entire viral life cycle, further securing the biosafety of this cell culture model. Based on this BSL-2 SARS-CoV-2 cell culture model, we developed a 96-well format high throughput screening for antivirals discovery. We identified salinomycin, tubeimoside I, monensin sodium, lycorine chloride and nigericin sodium as potent antivirals against SARS-CoV-2 infection. Collectively, we developed a convenient and efficient SARS-CoV-2 reverse genetics tool to dissect the virus life cycle under a BSL-2 condition. This powerful tool should accelerate our understanding of SARS-CoV-2 biology and its antiviral development.

## Author summary

The biosafety level-3 (BSL-3) classification of severe acute respiratory syndrome coronavirus 2 (SARS-CoV-2) impedes the research and antivirals development. We report a novel

**Funding:** Beijing Municipal Natural Science Foundation (M21001 to QD), Tsinghua University Initiative Scientific Research Program (2019Z06QCX10 to QD), National Natural Science Foundation of China (32041005 to QD), National Key Research and Development Program of China (2020YFA0707701 to RZ), Tsinghua-Peking University Center of Life Sciences (045-61020100120 to QD), Beijing Advanced Innovation Center for Structure Biology (100300001 to QD), Start-up Foundation of Tsinghua University (53332101319 to QD). The funders had no role in study design, data collection and analysis, decision to publish, or preparation of the manuscript.

**Competing interests:** I have read the journal's policy and the authors of this manuscript have the following competing interests: Q.D. and X.J. have filed a patent application on the use of the SARS-CoV-2 transcomplementation system and its use for anti-SARS-CoV-2 drug screening.

cell culture system for production of transcription and replication-competent SARS-CoV-2 virus-like-particles (trVLP) that can be used at BSL-2 laboratory for high-throughput neutralization and antiviral screening. This system consists of two components: a genomic viral RNA containing a deletion of nucleocapsid (N) gene, and a producer cell line expressing the N protein. The complete viral life cycle can be achieved and exclusively confined in the producer cells. Moreover, intein-mediated protein trans-splicing that splits N into two vectors further secures the biosafety of this system. Based on this system, we found residue-specific phosphorylation of N protein is critical for viral infection. Besides, high-throughput antiviral screening was developed and new drugs were discovered. Thus, this experimental system will facilitate the studies of SARS-CoV-2 biology and its antiviral development.

## Introduction

The coronavirus disease 2019 (COVID-19) caused by severe acute respiratory syndrome coronavirus 2 (SARS-CoV-2) is an ongoing pandemic[1]. As of 23 February 2021, more than 111 million cases of COVID-19 have been reported, resulting in more than 2.5 million deaths. Severe patients died of breathing difficulty to acute respiratory distress. Due to limited antiviral agents to combat SARS-CoV-2 infection, developing specific antiviral drugs against SARS-CoV-2 is still urgently needed[2].

SARS-CoV-2 belongs to the genus *Coronavirus*, the family *Coronaviridae*, and the order *Nidovirales*. Its genome is a single-stranded, positive-sense RNA with similar specific gene characteristics of known coronaviruses[3]. The viral genome encodes non-structural proteins, structural proteins and accessory proteins. The non-structural proteins carry all of the enzymatic activities important for viral replication. For example, the genome encodes an RNA-dependent RNA-polymerase complex (nsp7, nsp8 and nsp12), RNA capping machinery (nsp10, nsp13, nsp14 and 16) and additional enzymes such as proteases (the nsp3 PLpro and the nsp5 3CLpro) which cleave viral polyproteins[4,5]. Structural proteins include spike (S), envelope (E), membrane (M), and nucleocapsid (N) proteins[6]. The S, E and M proteins are embedded within the lipid envelope. The primary function of N protein is to package the ∼30 kb single stranded, 5′-capped positive-strand viral genome RNA into a ribonucleoprotein (RNP) complex. Ribonucleocapsid packaging is a fundamental part of viral self-assembly, and the RNP complex constitutes the essential template for replication by the RNA-dependent RNA polymerase complex[7]. In addition, the N protein has been shown to modulate the host antiviral response and may play regulatory roles in the viral life cycle [3]. The accessory proteins, encoded by ORF3a, ORF6, ORF7a, ORF7b, and ORF8 genes, are not directly involved in viral replication but interfere with the host innate immune response or are of unknown function[3,8,9].

The development of reverse genetics systems of coronavirus has profoundly advanced the study of this large-sized RNA virus. The cDNA of the coronavirus RNA genome is constructed using bacterial artificial chromosomes (BACs), *in vitro* ligation of CoV cDNA fragments, or vaccinia viral vector [10]. Recently, a SARS-CoV-2 full-length cDNA clone has been established using the *in vitro* ligation of cDNA fragments[11,12]. This system has been shown to be efficient for the recovery of infectious virus, and a reporter gene can be inserted into the viral genome to monitor virus replication, providing a good tool for high-throughput antiviral screening. However, experimentations involving live virus are restricted to BSL-3 laboratories, which hinders the study of SARS-CoV-2 and development of countermeasures. Therefore, it is

urgent to develop an efficient non-BSL-3 experimental system for SARS-CoV-2. Herein, we developed an N-based genetic complementation system to produce biologically contained, and transcription, replication-competent SARS-CoV-2 virus-like particles lacking N gene (SARS-CoV-2 ΔN trVLP). The lack of the viral N protein could be genetically complemented *in trans* by ectopic expression in packaging cells to produce the SARS-CoV-2 ΔN trVLP. SARS-CoV-2 ΔN trVLP could be propagated and passaged in the packaging cells while only results in single-round infection in wild-type cells. We applied this cell culture model for SARS-CoV-2 biology, antiviral evaluation and novel antivirals discovery.

## Results

### Design and assembly of SARS-CoV-2 GFP/ΔN genome

Nucleocapsid translated from a subgenomic RNA of SARS-CoV-2 has multiple functions and its primary function is participation in genomic RNA package and virus particle release. To test whether the function of N could be complemented *in trans*, we constructed SARS-CoV-2 GFP/ΔN genome, in which we replaced the regions encoding viral N (from nucleotides position 28274 to 29533) based on MN908947 genome with GFP reporter gene, and Caco-2 cells, an immortalized cell line of human colorectal adenocarcinoma cells, as packaging cell lines which stably express viral N protein by lentiviral transduction (**Fig 1**).

To assemble the molecular clone of SARS-CoV-2 GFP/ΔN genome, we utilized an *in vitro* ligation approach, which has been used for constructing the infectious clone of SARS-CoV-2 [11,12]. We divided the full-length cDNA of SARS-CoV-2 GFP/ΔN genome into a set of five fragments (A, B, C, D and E) and each fragment can be obtained by PCR using the chemically synthesized viral genome (MN908947 strain) as the template. Each DNA fragment was flanked by a type IIS restriction endonuclease site (BsaI or BsmBI) that ensures unidirectional assembly of intermediates into a full-length cDNA. In addition, we engineered a T7 DNA-dependent RNA polymerase promoter (T7 promoter) upstream of fragment A and a poly(A) tails at the downstream of fragment E, allowing for *in vitro* transcription of capped, polyadenylated transcript of viral genome (**Fig 1A**).

The five PCR-amplified DNA fragments were digested with BsaI or BsmBI to generate specific sticky ends (**Fig 1B**). The digested fragments were further purified and were ligated by T4 DNA ligase at 4˚C to generate the full-length cDNA of SARS-CoV-2 GFP/ΔN genome. The resulting 29.4-Kbp *in vitro* ligation products were confirmed by agarose gel electrophoresis (**Fig 1C**). Next, this *in vitro* ligation products were used as the template for *in vitro* transcription with the T7 RNA polymerase to generate the RNA transcript of SARS-CoV-2 GFP/ΔN genome (**Fig 1D**).

### Recovery and propagation of SARS-CoV-2 GFP/ΔN trVLP

Caco-2 cells are highly permissive for SARS-CoV-2 infection. As the recombinant SARS-CoV-2 GFP/ΔN virus like particles, lacking N gene, could potentially propagate in the cells supplied with viral N protein *in trans*, we established the Caco-2 cells stably expressing viral N gene by lentiviral transduction (designated as Caco-2-N cells). The expression of N was confirmed by flow cytometry and immunoblotting assay (**Figs 1E, S1A and S1B**).

Next, we sought to recover SARS-CoV-2 GFP/ΔN trVLP, the *in vitro* transcribed RNA transcript of viral genome was electroporated into Caco-2-N cells. Within 48h, GFP fluorescence can be readily observed, suggesting that viral genome replication and transcription occurs in the cell. After 96h, cytopathic effects (CPEs) were observed in the electroporated Caco-2-N cells, suggesting that the recombinant SARS-CoV-2 GFP/ΔN trVLP was produced and propagated (**S2 Fig**). We collected the cell culture supernatants (denoted as passage 0 (P0) virus),

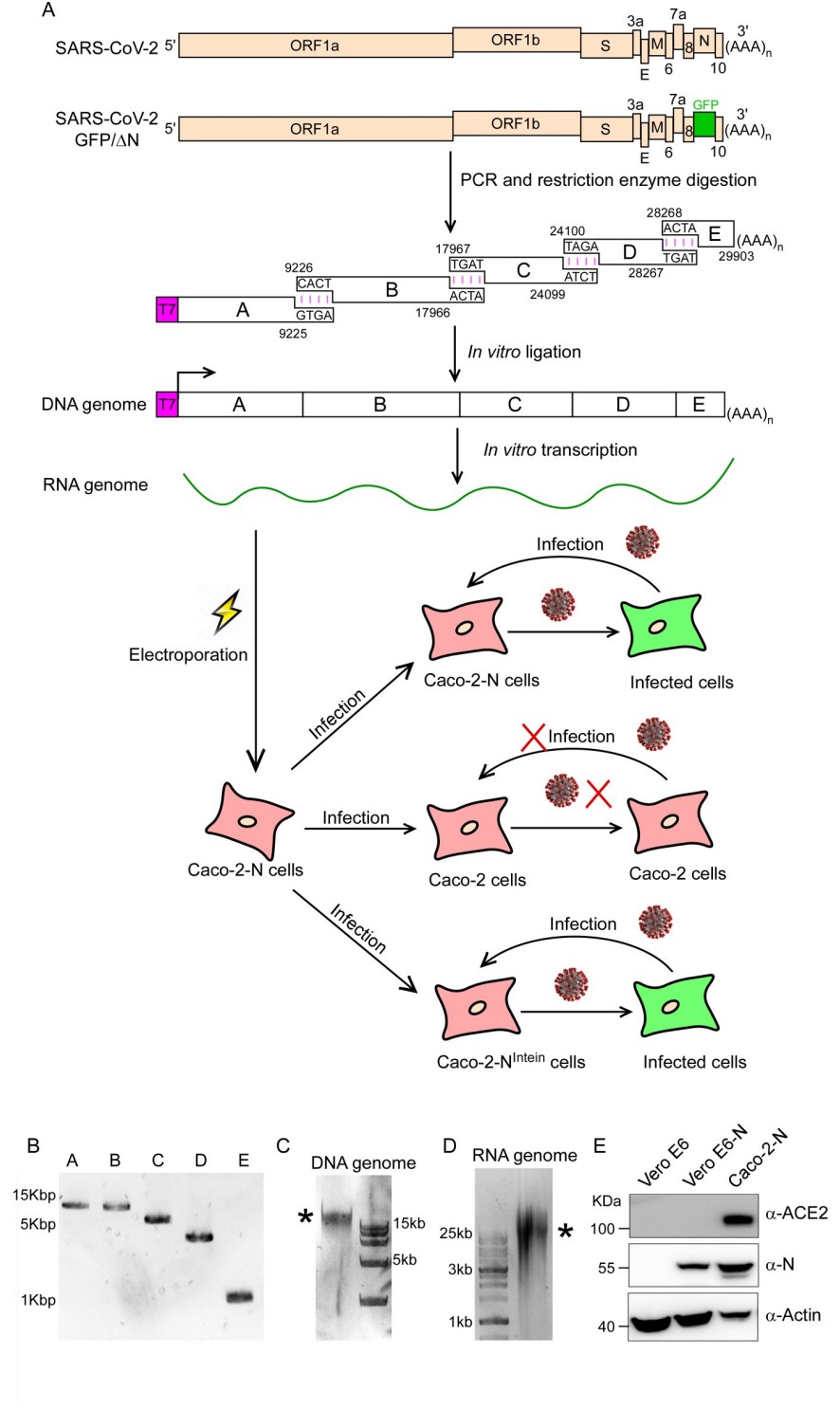

**Fig 1. Production of SARS-CoV-2 GFP/ΔN trVLP.** (A) The top rows show genetic organizations of the SARS-CoV-2 and SARS-CoV-2 GFP/ΔN genomes. The ORF of N is replaced with reporter gene (GFP here). The cDNA of SARS-CoV-2 GFP/ΔN genome was divided into five fragments designated as Fragment A, B, C, D and E, which could be obtained by PCR (B). Each cDNA fragment was flanked by a class IIS restriction endonuclease site (BsaI or BsmBI) and the nucleotide sequences and locations of the cohesive overhangs are indicated. The fragment cDNA were digested and purified for directed assembly of SARS-CoV-2 GFP/ΔN cDNA (see C panel, and the star indicates the genome-

length cDNA), which served as the template for *in vitro* transcription to generate viral RNA genome (see D panel, and the star indicates the genome-length RNA transcript). The viral genomic RNAs were electroporated into Caco-2-N cells. After 3 days, the supernatant was collected and inoculated with Caco-2 or Caco-2-N cells. (E). Western blotting assay was performed to detect the expression of N proteins and ACE2 in Caco-2-N cells, Vero E6 and Vero E6-N cells.

and inoculated them to Caco-2 or Caco-2-N cells (**Fig 2A**). GFP signal can be readily observed within 48 h, and further expanded within 72 h in Caco-2-N cells, whereas no signal was detected in Caco-2 cells (**Fig 2B**). Cells were collected for immunoblotting and RT-qPCR analysis at 72 h post-infection to detect viral spike antigen and RNA abundance. Consistent with the GFP expression, we could detect viral spike expression and high abundance of viral RNA in the Caco-2-N cells but not in Caco-2 cells (**Fig 2C and 2D**). RT-PCR analysis using a primer set outside the N-encoding region confirmed that the N gene was indeed replaced by GFP in the recombinant trVLP viral genome (**Fig 2E**).

To characterize the SARS-CoV-2 GFP/ΔN trVLP infection, two spike-specific mAbs (1F11 and 2F6)[13] were tested for their ability to neutralize infection of Caco-2-N cells. A neutralizing mAb specific for HIV gp120 (VRC-01) was also included as the control[14]. The mAbs were incubated with SARS-CoV-2 GFP/ΔN trVLP for 1 h at 37°C, and the trVLP–mAb mixtures were tested for infection of Caco-2-N cells, respectively. Viral infection was determined by flow cytometry at 48 h post-infection, and the results showed that 1F11 and 2F6 inhibited trVLP infection in a dose-dependent manner; in contrast, VRC01 had no effect on the trVLP infection (**Fig 2F**).

Soluble recombinant forms of the human ACE2 are able to bind SARS-CoV-2 spike protein and inhibit its interaction with cellular ACE2[15,16]. We therefore tested the ability of mouse IgG Fc fusion proteins of soluble human ACE2 (D30E) (hACE2 (D30E)-Fc)[16] to inhibit SARS-CoV-2 GFP/ΔN trVLP infection. F10scFv, an antibody specifically targeting HA of the Influenza A virus, was used as a negative control. The hACE2 (D30E)-Fc trVLP showed a dose-dependent neutralization of infectivity, inhibiting SARS-CoV-2 GFP/ΔN infection of Caco-2-N cells by 70% at 0.5μg/ml (**Fig 2G**). Together, these data demonstrated that the infection of SARS-CoV-2 GFP/ΔN trVLP recapitulates that of wild-type virus as its virus entry is also mediated by the interaction between viral spike and host ACE2.

## Characterization of the genetic stability of SARS-CoV-2 GFP/ΔN trVLP

Next, we sought to characterize the genetic stability of SARS-CoV-2 GFP/ΔN trVLP. For this purpose, we analyzed the rescued SARS-CoV-2 GFP/ΔN trVLP in Caco-2-N cells after 10 passages. The cell culture supernatants collected from SARS-CoV-2 GFP/ΔN RNA electroporated Caco-2-N cells were defined as P0, and the cell cultures collected from each subsequent passage on the Caco-2-N cells were defined as P1 to P10, respectively. The total RNAs extracted from each cell passage were used to perform RT-PCRs with the pair of primers to amplify the fragment between ORF8 and 3'UTR that covers the region of the inserted GFP reporter gene (**Fig 3A**). RT–PCR products of 1.5-Kbp and 1-Kbp were expected for WT genome and SARS-CoV-2 GFP/ΔN genome, respectively. SARS-CoV-2 GFP/ΔN trVLP was considerably stable for at least 3 serial passages since the 1-Kb RT–PCR products were detected at P3 trVLP (**Fig 3B**). The loss of GFP reporter gene was detected in the P4 trVLP as indicated by amplicon of < 1 Kb size (**Figs 3B and S3A**). No PCR product of greater than 1 Kb was detected in the all samples, suggesting that no heterologous RNA inserted into the SARS-CoV-2 GFP/ΔN genome, at least in the GFP report region.

To characterize the trVLP sequence variations in an unbiased manner, we performed deep sequencing analysis on the P1 and P10 trVLP genomes. The deep sequencing analysis provides

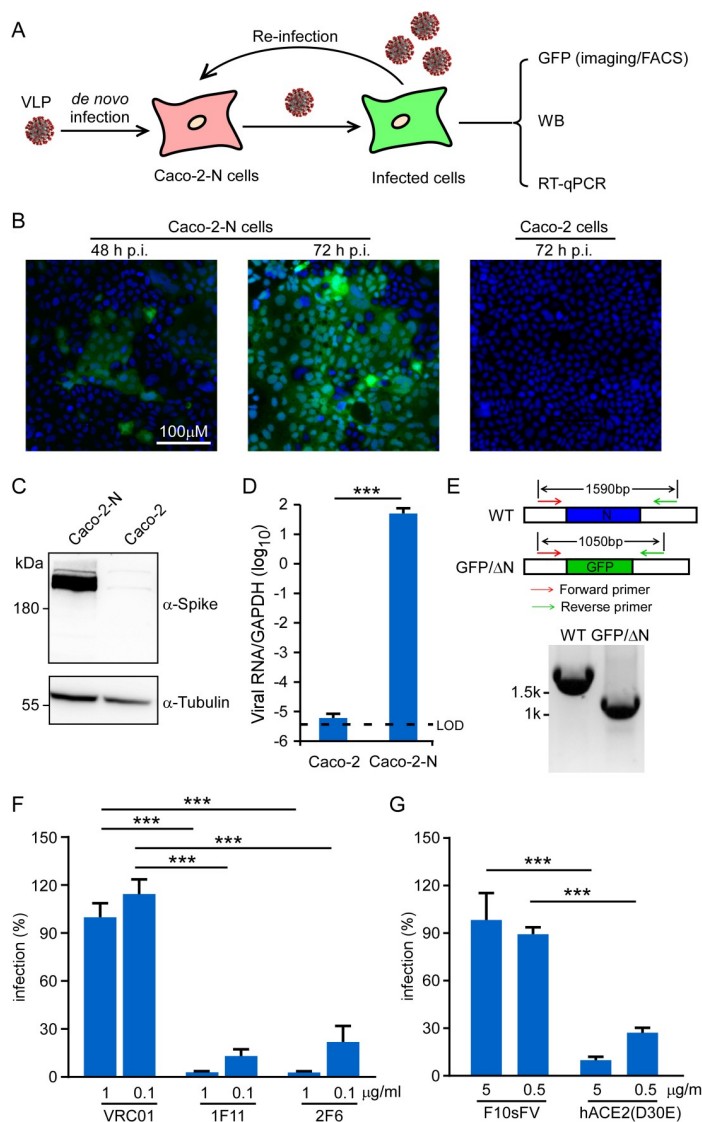

**Fig 2. The recombinant SARS-CoV-2-GFP/ΔN trVLP can propagate with the help of viral N protein.** (A) Experimental scheme. Caco-2 or Caco-2-N cells were infected with SARS-CoV-2 GFP/ΔN for 3h (MOI 0.05), washed, and incubated for an additional 72 h. GFP fluorescence were observed or quantified by microscopy or flow cytometry analysis. Viral RNA was determined by RT-qPCR assay; (B) GFP expression was observed in Caco-2 or Caco-2-N cells using microscopy at indicated time point after inoculation; Representative images from one of three independent experiments. (C) Cell lysates were resolved by SDS-PAGE and probed with anti-Spike and anti-Tubulin antibodies. Representative images from two independent experiments; (D) The total RNAs were extracted and RT-qPCR assays were conducted to determine viral RNA levels. Error bars represent the standard deviations from one of two independent experiments performed in triplicate; (E) RT-PCR analysis of the SARS-CoV-2 GFP/ΔN genome in Caco-2-N cells infected with recombinant virus using a primer set flanking the N region. The expected DNA sized were indicated in each genome, and DNA marker is shown on the left. Representative images from one of two independent experiments; (F-G) Recombinant SARS-CoV-2 GFP/ΔN virus was incubated with indicated doses of neutralizing mAbs against SARS-CoV-2 (1F11 and 2F6) or HIV (VCR01), as well as soluble human ACE2-Fc or F10sFV for 1 h prior to inoculation. The infection was analyzed by GFP expression 2 days later, and the number of positive cells was expressed as a percentage of that for the VRC01 or F10sFV treatment control. Error bars represent the standard deviations from three independent experiments (n = 6). ***, P < 0.001. Significance assessed by one-way ANOVA.

deep coverage, on the order of 30 million reads per sequencing sample (**S3B Fig**). Sequences of P1 or P10 were mapped to the SARS-CoV-2 and SARS-CoV-2 GFP/ΔN trVLP genomes, respectively (**Fig 3C and 3D**) and relative abundances of these sequences between P1 and P10,

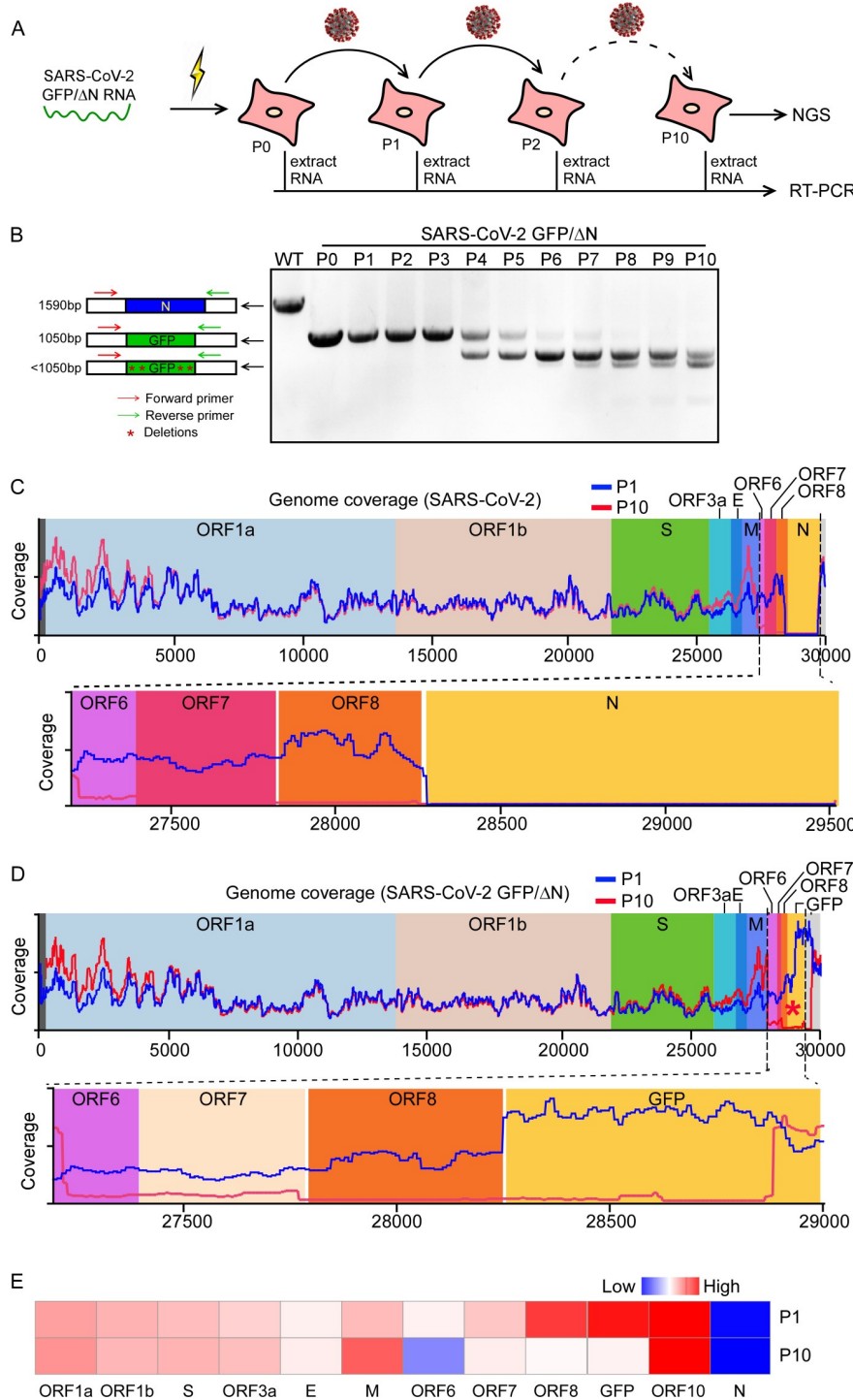

**Fig 3. Characterization of the genetic stability of SARS-CoV-2 GFP/ΔN trVLP.** (A) Detection of the GFP reporter gene during viral passage. RNAs were extracted from the VLP infected cells of P0 to P10 passage, respectively. (B) RT-PCR was performed with a primer pair flanking the N region of ORF8 and 3'UTR. The PCR products were resolved on an agarose gel using electrophoresis. The numbers of time points-samples-passage were denoted on the top of each lane. Representative images from one of three independent experiments; (C-D) RNA-seq coverage of virus derived reads aligned to SARS-CoV-2 (C) or SARS-CoV-2 GFP/ΔN (D) genome, respectively. (E) Heatmap shows the expression levels of each subgenomic RNA of P1 or P10 trVLP.

were also compared (**Figs 3E and S3C**). The deep sequencing analysis could not detect N sequences in the both P0 and P10 genomes (**Fig 3C**), and GFP sequences were readily detected in the P1 genome with high abundance, however, it was rarely detected in the P10 genome (**Fig 3D**), due to GFP sequences deletion (**Figs 3B and S3A**). Additionally, we found that the subgenomic RNAs of ORF6, ORF7 and ORF8 were dramatically decreased in the P10-trVLP infected cells compared with that of P1 VLP (**Fig 3E**), indicating that ORF6, ORF7 and ORF8 might be dispensable for SARS-CoV-2 after its adaptation into Caco-2 *in vitro*, this observation is consistent with other reports that deletions of these regions were observed in clinical samples by deep sequencing analysis[17–20].

## Reconstitution of functional N protein by split intein-mediated protein ligation

Inteins are intervening protein sequences within a host protein that mediate their self-excision from the precursor protein and ligates the flanking N- and C-terminal fragments (exteins)[21]. Split inteins are a subset of inteins that are expressed as two separate polypeptides at the ends of two host proteins and catalyze their trans-splicing, resulting in the generation of a single larger polypeptide (**Fig 4A**). To further minimize the chance of recombination of N into the SARS-CoV-2 GFP/ΔN genome, we aimed to split the N gene into two separate elements using a naturally split intein embedded within the catalytic subunit of DNA polymerase III (DnaE) in many species of cyanobacteria (Npu intein)[22]. Npu intein activity is context-dependent, and Cys as first residue in the C-extein is required for efficient trans-splicing. However, there is no Cys residue in SARS-CoV-2 N protein. In order to split the N, we had to find the appropriate splice sites that would have two well-folded, yet stable protein fragments, and also substitute first residue in the C-extein with Cys without disruption of N protein function. To locate the splice sites according to these requirements we chose three splice sites in the N protein, to have the N-intein A152C, S176C and G212C (**Fig 4A**). As for each of N-intein above, we constructed two lentivirus vectors encoding either the N- or the C-terminal half of the N protein fused to the N- and C-terminal halves of the Npu intein, having $N^N$-$Int^N$ and $Int^C$-$N^C$, respectively (**Fig 4A**). Each lentivirus vector included appropriate regulatory elements (promoter and a polyadenylation signal) and a Flag tag to allow detection of the full-length reconstituted N protein (**Fig 4A**). We then transduced $N^N$-$Int^N$ and $Int^C$-$N^C$ either individually or together in Caco-2 cells, and the full-length N protein reconstitution was assessed by Western blotting assay. We could not detect splicing above negligible levels of N protein by N-intein (A152C) (**lane 2, Fig 4B**), while N-intein (S176C) and N-intein (G212C) could reconstitute into full-length N protein with S176C or G212C point mutation, respectively (**lane 5 and 8, Fig 4B**). Next, recombinant SARS-CoV-2 GFP/ΔN trVLP (P1) was inoculated to Caco-2 cells transduced with N-intein as indicated, and GFP fluorescence was detected after two days only in the cells transduced either with a single lentivirus that encodes full-length N or with the combination of N-intein (G212C), but not in the cells with the single N- and C-terminal N-intein (G212C). As expected, GFP fluorescence was also not detected in cells transduced with N-intein (A152C), of which the splicing did not occur; interestingly, N-intein (S176C) could ligate a full-length N (S176C), but fails to support virus infection, suggesting that the S176C mutation probably impairs N protein function (**Fig 4C and 4D**). Consistent with the GFP expression, the subgenomic RNA of E can be readily detected in cells transduced either with a single lentivirus that encodes full-length N or with the combination of N-intein (G212C), but not others (**Fig 4E**). Together, we showed that the N-intein (G212C) was capable of efficiently trans-splicing to generate a functional N (G212C) protein to support SARS-CoV-2 GFP/ΔN trVLP infection. As the N-intein was split into separate constructs, it would further reduce the

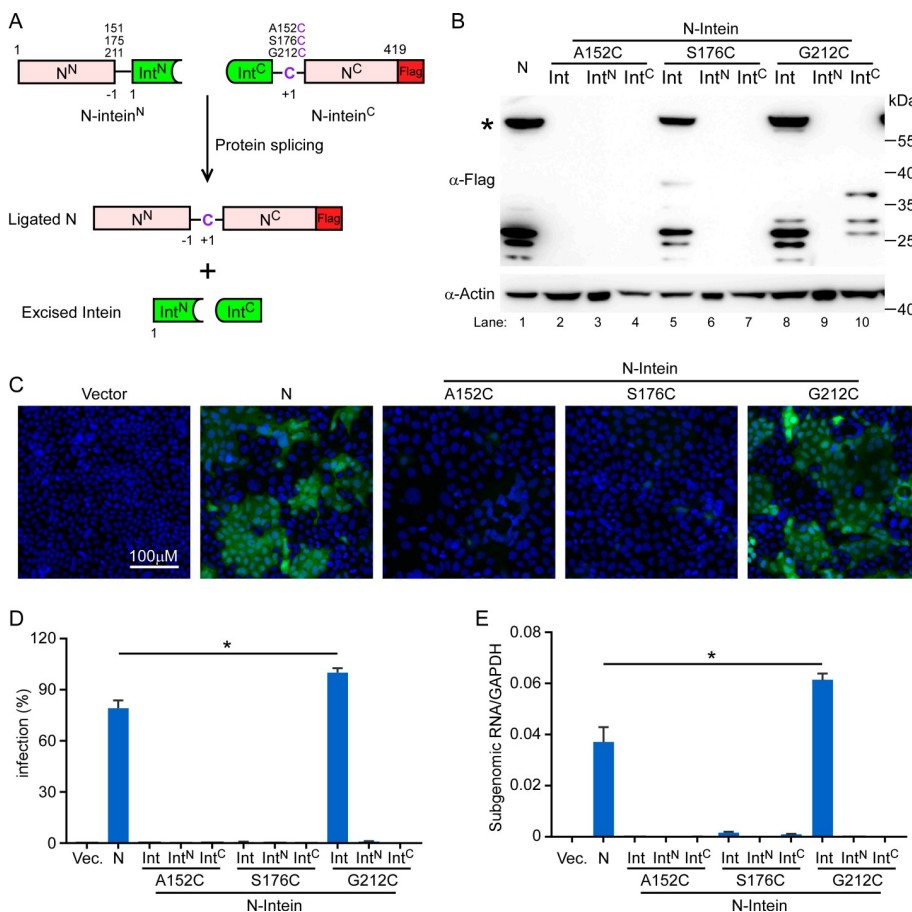

**Fig 4. Reconstitution of functional N protein by intein-mediated protein splicing.** (A) Scheme depicting of intein-mediated protein trans-splicing to reconstitute full length N protein. (B) Western blot (WB) analysis of lysates from Caco-2 cells transduced with either full-length N or intein-N lentiviruses. The star indicates the full-length N protein. The WB is representative of three independent experiments. (C) GFP fluorescence in Caco-2-N cells infected cell culture medium (containing SARS-CoV-2 GFP/ΔN progeny) collected from each Caco-2-N$^{int}$ cells which was inoculated with SARS-CoV-2 GFP/ΔN trVLP at 2 days of culture. The image is representative of n = 4. (D) Cells were harvested to quantify GFP expression by flow cytometry analysis, and (E) Subgenomic RNA of E were determined by RT-qPCR assay. Error bars represent the standard deviations from one of three independent experiments performed in triplicate. *, P < 0.05. Significance assessed by one-way ANOVA.

potential biosafety concerns of this SARS-CoV-2 GFP/ΔN trVLP cell culture model. Notably, G212C is closed to the incidences of R203K/G204R[23], and it is required to be further validated the ligation and function of N (R203K/G204R)-intein (G212C).

## Residue-specific phosphorylation of N protein is critical for viral infectivity

Coronavirus N protein is an extensively phosphorylated, highly basic, vital structural protein and the primary function of which is to form a helical ribonucleoprotein complex with viral RNA (RNP) as core structure of the virion. A variety of other functions have been ascribed, such as viral genome transcription and replication, or evasion of antiviral immunity. SARS-CoV-2 N protein is highly homologous to the N protein of SARS-CoV, with 91% identity, while exhibited 48% identity with that of MERS-CoV (**Fig 5A**). Several proteomics profiling analyses have been performed and reveal that N protein of SARS-CoV-2 is extensively phosphorylated at multiple sites (**Figs 5A and S4**). However, the roles of N protein phosphorylation

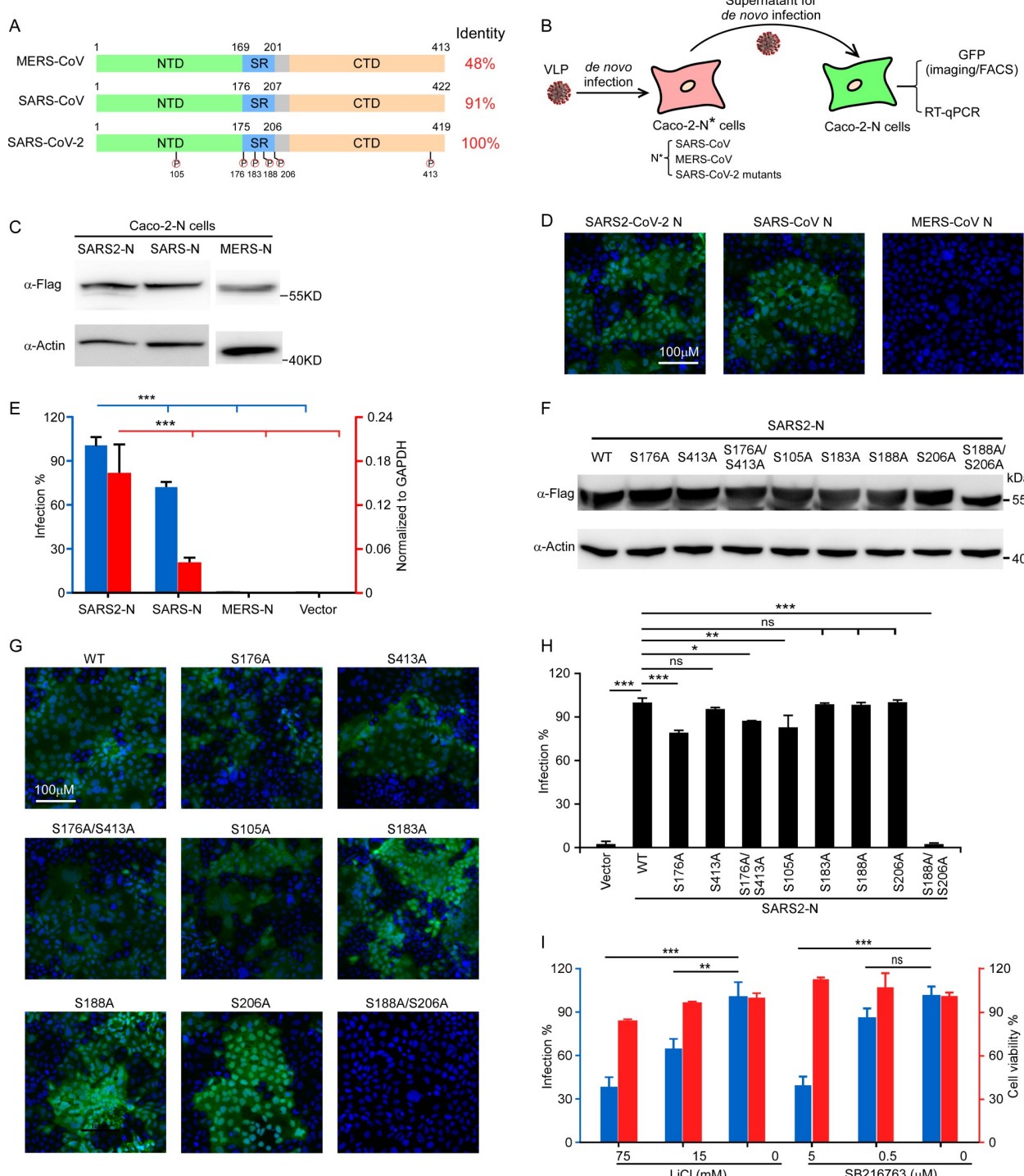

**Fig 5. Site-specific phosphorylation of N is required to support virus life cycle.** (A) Schematics and alignments of N proteins from MERS-CoV, SARS-CoV and SARS-CoV-2. The phosphorylation sites in SARS-CoV-2 N protein were highlighted. (B) Schematic presentation of assessment of N variants function. The trVLP inoculated with Caco-2 cells transduced with N variants, and the cell culture medium were collected to infect the Caco-2-N cells, and GFP expression analyzed by flow cytometry/microscopy or viral subgenomic RNA abundance were determined by RT-qPCR. (C) Western blotting assay was performed to detect the N proteins expression in Caco-2 cells transduced with distinct N genes from SARS-CoV-2, SARS-CoV or MERS-CoV. (D-E) The cell culture medium was collected from SARS-CoV-2 GFP/ΔN trVLP infected Caco-2 cells expressing N from SARS-CoV-2, SARS-CoV or MERS-CoV to infect the naïve Caco-2-N cells. GFP were observed using microscopy and cellular RNA was extracted for RT-qPCR analysis to determine viral subgenomic RNA levels. (F) Western blotting assay detected the expression of SARS-CoV-2 N WT or mutants in Caco-2 cells. (G-H) The cell culture medium was collected from SARS-CoV-2 GFP/ΔN trVLP infected Caco-2 cells expressing

SARS-CoV-2 N mutants to infect the naïve Caco-2-N cells. GFP were observed using microscopy and cellular RNA was extracted for RT-qPCR analysis to determine viral subgenomic RNA levels. (I) GSK-3 inhibitors LiCl or SB216763 treated Caco-2-N cells inoculated with SARS-CoV-2 GFP/ΔN trVLP, the cell culture medium was then inoculated with Caco-2-N cells. RNA was extracted for RT-qPCR analysis to determine viral subgenomic RNA levels. Cell viability was evaluated by CellTiter-Glo assay. Error bars (E, H and I) represent the standard deviations from one of three independent experiments performed in triplicate. n.s. no significance; *, P < 0.05; **, P < 0.01; ***, P < 0.001. Significance assessed by one-way ANOVA.

remain unclear. Our N-based genetic trans-complemented cell culture model offers an opportunity to specifically study N protein function in viral life cycle. Firstly, we determined whether SARS-CoV-2 GFP/ΔN trVLP infection can be complemented by N proteins from different coronavirues. We used SARS-CoV-2 GFP/ΔN trVLP to infect the Caco-2 cells transduced with N from SARS-CoV-2, SARS-CoV or MERS-CoV. Two days later, the cell culture supernatants from each cells were collected to infect Caco-2 cells transduced with SARS-CoV-2N (Caco-2-N cells as previously used) to test whether SARS-CoV-2 GFP/ΔN trVLP were assembled in the Caco-2 cells transduced with distinct N proteins. Two days later, the Caco-2-N cells were collected and GFP or viral RNA was quantified by flow cytometry or RT-qPCR, respectively (**Fig 5B and 5C**). SARS-CoV N protein with 91% identity with that of SARS-CoV-2, but not MERS-CoV N protein with 48% identity with that of SARS-CoV-2, could rescue SARS-CoV-2 GFP/ΔN trVLP (**Fig 5D and 5E**), suggesting that coronavirus N protein has virus-specific mechanism to recognize viral genome to achieve its function, meanwhile, N proteins from SARS-CoV and SARS-CoV-2, with high genetic similarity, have redundant function to some degree.

As SARS-CoV-2 N is heavily phosphorylated at multiple sites especially within the central Ser-Arg (SR)-rich motif, we are interested in the roles of phosphorylation in N function. For this purpose, we mutated S176, S413, S176/413, S105, S183, S188, S206, S188/206 as the conservation of these residues with that of SARS-CoV into alanine to specifically dissect their function. Notably, GSK-3 is the kinase responsible for the phosphorylation of this SR-rich motif in SARS-CoV N protein, which are primed by the phosphorylation of Ser-189 and Ser-207 (Ser-188 and Ser-206 in SARS-CoV-2 N protein accordingly)[24,25]. We generated the Caco-2 cells lentivirally transduced with the N variants as indicated. As shown in Western blotting assay, the mutations did not alter the protein expression and stability in the Caco-2 cell (**Fig 5F**), We noted that N with the S188A/S206A double mutations migrated slightly faster than WT and other mutants, probably because blockade of the initial priming phosphorylation would prohibit subsequent phosphorylation events by GSK-3, which was observed in SARS-CoV [24]. Next, we inoculated the Caco-2 cells expressing different N variants with SARS-CoV-2 GFP/ΔN trVLP, and cell culture supernatant was collected 48 h later to infect the naïve Caco-2-N cells, and cells were collected to observe or determine GFP expressing by microscopy or flow cytometry 2 days later. Interestingly, most of the phosphorylation null mutants were able to assemble virus-like particles with comparable or slightly reduced efficiencies than WT. However, S188A/S206A double mutations completely abolished N function (**Fig 5G and 5H**), highlighting the critical role of S188 and S206 for N function.

To further investigate whether GSK-3 contributing N protein phosphorylation to regulate virus life cycle, we treated Caco-2-N cells with LiCl or SB216763, which are specific inhibitors of GSK-3 and inoculated cells with SARS-CoV-2 trVLP spontaneously. Two days later, cell culture medium was collected and infect Caco-2-N cells for additional 2 days, and then cells were harvested for flow cytometry analysis of GFP expression. As expected, the LiCl or SB216763 could inhibit GFP expression in a dose-dependent manner, indicating that inhibition of GSK-3 could block N phosphorylation, thus impairing SARS-CoV-2 trVLP production. Given the

vital role of the N protein in multiple stages of the viral life cycle, inhibition of N functions by modulating host cell kinases may be viable strategies for combating SARS-CoV-2 infections.

## Evaluation of the antivirals using SARS-CoV-2 GFP/ΔN VLP cell culture model

To test the utility of this system in anti-viral drug screening, we evaluated the efficacy of IFN-β, remdesivir, GC376, lopinavir, and ritonavir in inhibiting SARS-CoV-2 GFP/ΔN trVLP infection. Caco-2-N-intein (G212C) cells were treated with IFN-β with 0.2–20 pg/ml for eight hours prior to infection. Then cells were infected with SARS-CoV-2 GFP/ΔN trVLP at a multiplicity of infection (MOI) of 0.05. After 48 h, the cells were collected and GFP fluorescence, the proxy of virus infection, was quantified by flow cytometry analysis. Remarkably, even at 0.2 pg/ml IFN-β we observed 60% reduction of the GFP fluorescence (Fig 6A). This is consistent

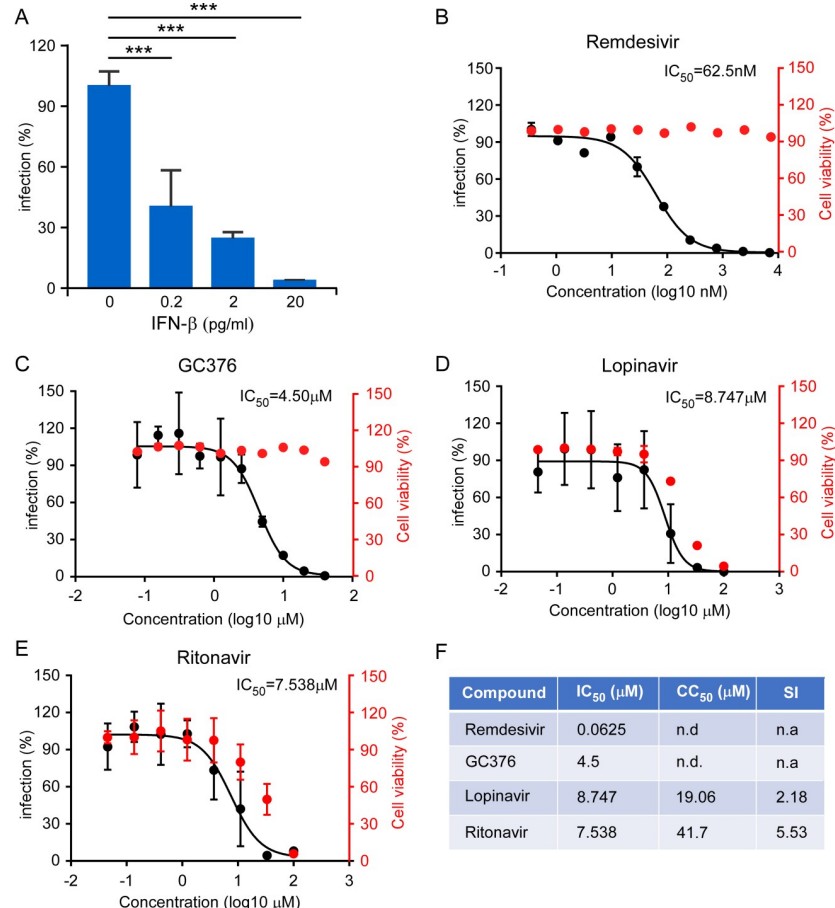

**Fig 6. Inhibition of recombinant SARS-CoV-2 GFP/ΔN trVLP infection by IFN and antivirals.** (A) IFN-β pretreated Caco-2-N[int] cells were subsequently infected with trVLP and cells were subjected to flow cytometry analysis for quantify the GFP fluorescence at 2 days post-infection. Error bars represent the standard deviations from three independent experiments (n = 6). (B-E) Antiviral effect of remdesivir, GC376, lopinavir and ritonavir. The drug treated cells were infected with trVLP and GFP fluorescence was quantified at 48h after infection. The cytotoxic effect of each drug at indicated concentrations were determined by CellTiter-Glo cell viability assay. The virus infection or cytotoxicity is plotted versus compound concentration (n = 3 biological replicates for all compounds). The black dots indicate replicate measurements, and the black lines indicate dose-response curve fits. The red dots indicate cytotoxicity. IC$_{50}$ values were calculated using Prism software and is representative of one of three independent experiments performed in triplicate. Three independent experiments had similar results. (F) Comparison of antiviral activity and cytotoxicity of each compound. Selectivity Index (SI), a ratio that compares a drug's cytotoxicity and antiviral activity was also calculated. n.d. = not detected; n.a. = not applicable.

with recent reports that SARS-CoV-2 is sensitive to type I interferon treatment[11,26–28]. Remdesivir and GC376, which target virus RNA dependent RNA polymerase (RdRp) and 3CLpro respectively, have been reported to be potent antivirals against SARS-CoV-2[29–33]. Lopinavir and ritonavir-HIV protease inhibitor, is a combination antiviral medicine used to treat HIV[34], which could inhibit SARS-CoV and MERS-CoV infection *in vitro*, and they may target SARS-CoV-2 Nsp5 (3CLpro) to inhibit virus infection. To test potential dose-dependent antiviral activity of those drugs in our system, we incubated Caco-2-N-intein (G212C) cells with various concentrations of those drugs and simultaneously infected the cells with SARS-CoV-2 GFP/ΔN trVLP at a MOI of 0.05. After 2 days, GFP fluorescence was determined (**Fig 6B-6E**). Remdesivir and GC376 exhibited potent antiviral effect with $IC_{50}$ = 62.5 nM and 4.5 μM respectively, with essentially no apparent cytotoxic effect (**Fig 6B and 6C**). In contrast, Lopinavir or ritonavir inhibited SARS-CoV-2-GFP/ΔN trVLP with $IC_{50}$ = 8.7 μM, or 7.7 μM, while those drugs both show serious cytotoxicity at the $IC_{50}$ concentration (**Fig 6D and 6E**), compromising their clinical utilities, which is in line with the fact that lopinavir and ritonavir as no significant beneficial effect was observed in a randomized trial established in March 2020 with a total of 1,596 patients[35].

These results demonstrated that our experimental system can be used for evaluation of antivirals and could be potentially developed for high-throughput screening of antiviral compounds.

## Identification of potent antivirals against SARS-CoV-2 virus using trVLP cell culture model by high-throughput screening

To provide proof-of-concept that our system could be utilized in high-throughput screening, we performed HTS of Topscience natural product library containing 377 drugs (**Fig 7A**) and the potential hit compounds were further assessed using authentic SARS-CoV-2 to confirm the antiviral activities *in vitro*. DMSO or remdesivir were included as the negative or positive control.

Among the 377 compounds of the compound library, 10 hit molecules showed equal or higher inhibition with an inhibitory efficiency ≥ 60% (**Fig 7A**). In addition, we excluded five hits due to the visible cytotoxicity. This criterion allowed the selection of five hits as the highest confident hits: salinomycin, tubeimoside I, monensin sodium, lycorine chloride and nigericin sodium (**Fig 7A**). Among these five compounds, lycorine chloride, salinomycin and monensin sodium inhibit HCoV-OC43 infection as previously reported[36] and monensin sodium blocks avian infectious bronchitis virus (IBV) infection[37]. Notably, a recent study demonstrated that salinomycin possessed a potent antiviral activity to inhibit SARS-CoV-2 infection *in vitro*[38], which further demonstrated that our system could be used for HTP antiviral screening. We next determined the $IC_{50}$ of the hit compounds using authentic SARS-CoV-2 virus. Salinomycin showed SARS-CoV-2 antiviral activity with an $IC_{50}$ and $CC_{50}$ of 2.836 and 20.23 μM, respectively, and selectivity index (SI = $CC_{50}/IC_{50}$) of 7.13. In comparison, other four compounds did not show dramatic cytotoxic effect in the tested concentrations. Of note, tubeimosde I exhibited an $IC_{50}$ of 1.371 μM; monensin sodium exhibited an $IC_{50}$ of 0.632 μM; lycorine chloride showed antiviral activity with an $IC_{50}$ of 0.773 μM, and nigericin sodium, exhibited an $IC_{50}$ of 11.25 μM. These results demonstrated that the compounds we identified using SARS-CoV-2 GFP/ΔN trVLP system exhibited potent antiviral activity against authentic SARS-CoV-2 infection, and our screening provided new candidate compounds to effectively treat infection of SARS-CoV-2.

## Discussion

As its high pathogenicity and the lack of effective therapeutics, SARS-CoV-2 is classified as a biological safety level 3 (BSL-3) pathogen[39], which has hindered the drug discovery and biological research due to biocontainment requirements. In this study, we developed an *in vitro*

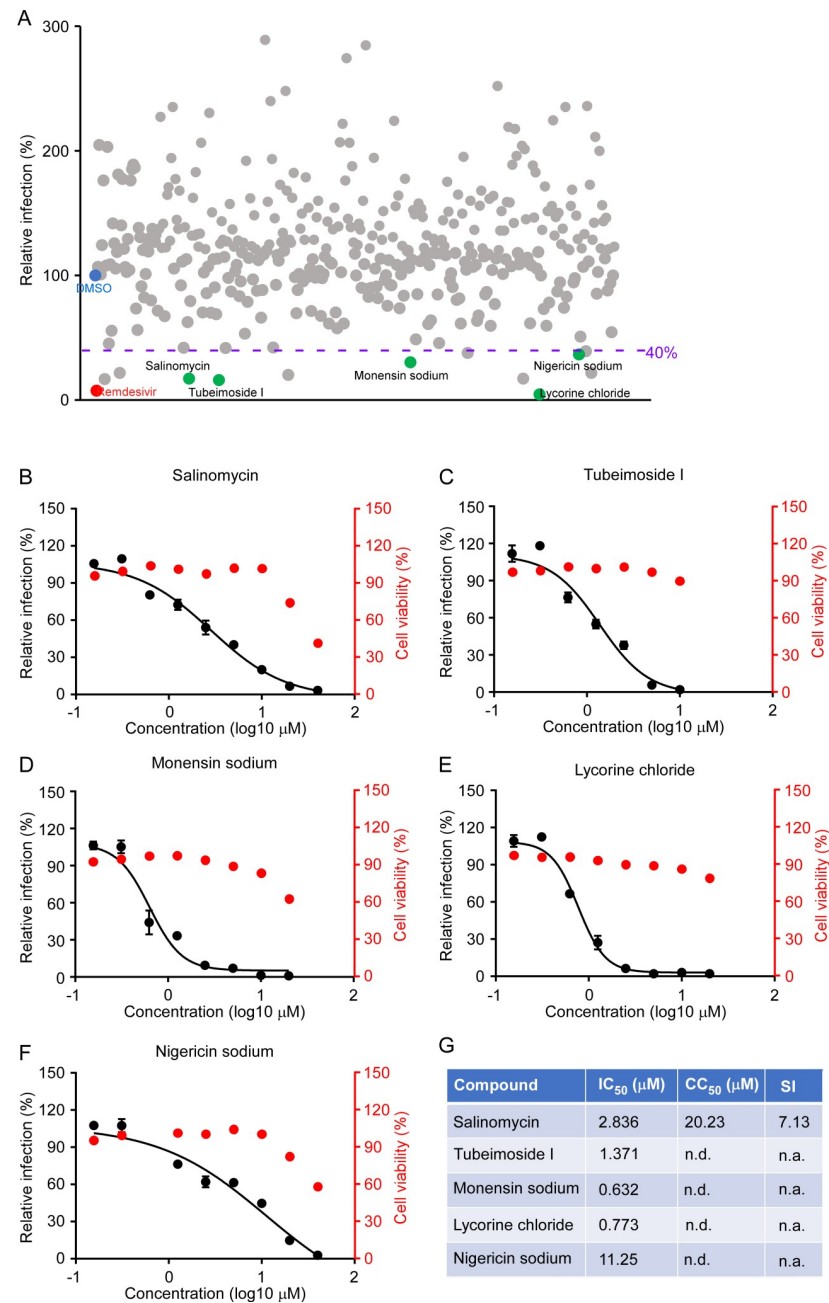

**Fig 7. High throughput screening of antivirals against SARS-CoV-2 infection using trVLP system.** (A) Screening of 377 compounds from Topscience Natural Product Library and hits selection. The purple dot line represents the threshold (40%) for positive hit compounds. DMSO (blue) and remdesivir (red) are used as the control for the screening. Each dot represents a single compound, and the green dots represent the promising candidates which exhibited potent antiviral activity without dramatic cytotoxic effect. (B-F) Dose response curves of selected hit compounds. Compounds concentrations are presented in log scale for logarithmic interpolation. Dose response curves were generated using GraphPad Prism software version 7.0. $IC_{50}$ values were calculated using Prism software and is representative of one of three independent experiments. Error bars represent the standard deviations from one of three independent experiments performed in triplicate. (G) Comparison of antiviral activity and cytotoxicity of each compound. Selectivity Index (SI), a ratio that compares a drug's cytotoxicity and antiviral activity was also calculated. n.d. = not detected; n.a. = not applicable.

cell culture system to produce the recombinant SARS-CoV-2 virus lacking the N-encoding region in the viral genome (SARS-CoV-2 ΔN). Recombinant SARS-CoV-2 ΔN virus can expand and propagate in packaging cells (Caco-2-N) but results in only single-cycle infection in naïve Caco-2 cells, which biologically contained the virus in the cells expressing N protein. This BSL-2 SARS-CoV-2 possesses a reporter gene GFP, providing a surrogate readout for authentic viral infection. We monitored the recombinant virus infection in the Caco-2-N cells for one month and NGS sequencing result suggested that no recombination was detected. In addition, we utilized the split intein-mediated protein ligation to reconstitute N function which further ensure the biosafety of this system. However, we have not performed the recombination analysis in Caco-2-N or Caco-2-N^intein cells co-infection of SARS-CoV-2 trVLP with WT SARS-CoV-2 or other related beta-coronaviruses, which could potentially increase the risk of the recombination and need further investigation.

This cell model represents a unique system in the basic research application for better understanding SARS-CoV-2 life cycle. Virus has evolved since its outbreak in the end of 2019, and some mutations or deletions have been observed. However, the functional consequences of these mutations or deletions on virus infectivity or pathogenesis are poorly characterized. Herein, we utilized our model system to study the roles of N in the SARS-CoV-2 life cycle. Since N can be expressed alone *in trans*, it is convenient to perform mutagenesis on N to dissect its detailed function. Moreover, the introduction of mutations *in trans*-expressed N will avoid the *cis* effects of the mutations, for example, the disruption of critical RNA secondary or tertiary structures in the SARS-CoV-2 genome, thereby providing a more appropriate system to specifically evaluate the biological roles of domains, motifs, or amino acid residues within the N protein. Additionally, we inserted a Flag tag at the C terminus of N, which did not impair the ability of N to rescue viral production. With this Flag tag, N can be detected and immunoprecipitated by an anti-Flag antibody (**S4A Fig**). Multiple amino acids in N protein can be phosphorylated, but our data demonstrated that most of these phosphorylation may not be required for N function at least *in vitro*. Meanwhile, we also identified numerous host factors associated with N protein (**S4A Fig** and **S2 Table**), notably, we also found that N protein could interact with G3BP1 and G3BP2, the stress granule assembly proteins, which was in line with previous studies[40–42]. Recent studies found that N protein could impair the stress granule assembly to escape the antiviral effect [41,43]. Thus, the trVLP system provides a new tool to study host factors and viral proteins that may interact with N during SARS-CoV-2 infection.

Development of effective therapeutics for COVID-19 remains an urgently unmet medical need. This recombinant trVLP recapitulates the complete SARS-CoV-2 life cycle in the Caco-2-N or Caco-2-N^intein cells. The reporter readout of the virus, such as fluorescent proteins or luminescent proteins, offers a rapid, real-time, quantitative and less labor-intensive measure than traditional methods of viral titer reduction assay. Importantly, the reporter virus-based assay could cooperate with a BSL-2 compatible high-content screening platform to facilitate antiviral screening. Thus, we developed a 96-well format to screen the antiviral compounds in the Topscience Natural Compounds Library, and we identified five compounds which could efficiently block SARS-CoV-2 infection. Among them, lycorine, salinomycin and monensin have been reported as the potent inhibitors against HCoV-OC43 infection[36], and salinomycin could block SARS-CoV-2 infection as reported recently[38]. Those data further validate the suitability of our trVLP system in drug discovery. In our screening, we identified Tubeimoside I and nigericin sodium as novel compounds which exhibited potent antiviral activities against authentic SARS-CoV-2 infection *in vitro*. Future studies could be performed to evaluate their antiviral activities *in vivo*.

Additionally, there is an urgently need for effective vaccines to contain SARS-CoV-2 pandemic[39]. The recombinant SARS-CoV-2 lacking of N gene should provide a new means of vaccine development. The greatest advantage of SARS-CoV-2 ΔN is that this virus possesses all the structural viral proteins to induce humoral immune responses and that, upon infection, it could produce all the nonstructural viral proteins in host cells to induce cell-mediated immune responses. Of course, further studies, especially in animals, are needed to determine the immunogenicity, safety, and efficacy of it.

In summary, the biologically contained SARS-CoV-2 trVLP lacking the N gene represents a safe, alternative experimental system to study SARS-CoV-2 biology and to screen antiviral compounds and this novel system will greatly accelerate current SARS-CoV-2 research efforts.

## Materials and methods

### Cell culture

HEK293T, Vero, Vero E6, A549 and Caco-2 cells were maintained in Dulbecco's modified Eagle medium (DMEM) (Gibco, China) supplemented with 10% (vol/vol) fetal bovine serum (FBS), and 50 IU/ml penicillin/streptomycin in a humidified 5% (vol/vol) $CO_2$ incubator at 37˚C. All cell lines were tested negative for mycoplasma.

### Cloning of the SARS-CoV-2 GFP/ΔN cDNA

cDNAs (Wuhan-Hu-1, MN908947) of SARS-CoV-2 GFP/ΔN were synthesized from the GenScript company (the N gene is replaced with gene of green fluorescent protein (GFP)). PCR was conducted to amplify fragments A, B, C, D and E using high fidelity PrimeSTAR Max DNA Polymerase (Takara). T7 promoter was introduced upstream of 5' UTR of SARS-CoV-2 genome in fragment A. To guarantee a seamless assembly of the full-length cDNA, type IIS restriction endonuclease sites (BsaI or BsmBI) were introduced at both ends of PCR fragments. The primers used for the PCR assay were listed in **S1 Table**.

### Assembly of a Full-Length SARS-CoV-2 GFP/ΔN cDNA

PCR fragments were digested with BsaI or BsmBI restriction enzyme (NEB) to get specific sticky end. Digested fragments are purified by E.Z.N.A gel extraction kit (Omega). Fragment A, B are ligated first by T4 DNA ligase (NEB) in 40μl system. At the same time, fragments C, D, E are also ligated in another tube at 4˚C for 24 hours. Then, fragment A, B and C, D, E are combined together added with 2μl T4 DNA ligase buffer and 2μl T4 DNA ligase to 100μl at 4˚C for another 24 hours. At the end of ligation, we took 5μl product to run an agarose gel to check the efficiency of ligation. Full-length assembly cDNA was phenol/chloroform extracted, isopropanol precipitated, and resuspended in 10μL nuclease-free water. Requests for resources and reagents relevant with this study should be directed to corresponding author Dr. Qiang Ding (qding@tsinghua.edu.cn).

### RNA *in vitro* transcription, electroporation and virus production

RNA transcript was *in vitro* transcribed by the mMESSAGE mMACHINE T7 Transcription Kit (ThermoFisher Scientific) in 30μl system with some modifications. Twenty micrograms of viral RNA and 20μg N mRNA were mixed and added to a 4-mm cuvette containing 0.4 mL of Caco-2-N cells ($8\times10^6$) in Opti-MEM. Single electrical pulse was given with a GenePulser apparatus (Bio-Rad) with setting of 270V at 950μF. GFP signal can be observed 17 hours post electroporation. Three days post electroporation, P0 virus was collected and Caco-2-N cells were infected with P0 virus to amplify virus.

## Lentivirus packaging

Vesicular stomatitis virus G protein (VSV-G) pseudotyped lentiviruses were produced by transient cotransfection of the third-generation packaging plasmids pMD2G (catalog no. 12259; Addgene), psPAX2 (catalog number 12260; Addgene) and the transfer vector pLVX by Vigo-Fect DNA transfection reagent (Vigorous) into HEK293T cells. The medium was changed 12 h post transfection. Supernatants were collected at 36, 60 and 84 h after transfection, pooled, passed through a 0.45-μm filter, aliquoted, and frozen at -80˚C refrigerator.

## RNA isolation and RT-qPCR

Total cellular RNA was isolated using TRIzol reagent (Thermo, 15596018). To analyze the RNA level of SARS-CoV-2 in infected cells, quantitative real-time PCR was performed. In brief, 1μg total RNA was reverse transcribed using ReverTra Ace qPCR RT Kit (TOYOBO, FSQ-101) to produce cDNA with random primers. Reactions of qPCR were carried out using the 2×RealStar Green Power Mixture (Genstar, A311) according to the instruction. The qPCR primers for viral RNA were as follows: THU-2190 (5'- CGAAAGGTAAGATGGAGAGCC-3') and THU-2191 (5'- TGTTGACGTGCCTCTGATAAG-3'). The sequences of the qPCR primers for GAPDH was described previously[44]. Relative expression levels of the target genes were calculated using the comparative cycle threshold (CT) method. All data were normalized relative to the housekeeping gene GAPDH.

## RNA-seq and data analysis

Total RNA was extracted by using TRIzol Reagent (Invitrogen) according to the manufacturer's protocol. The rRNAs were removed by using Ribo-Zero Gold module of Illumina TruSeq stranded total RNA library prep kit (RS-122-2201) and then cDNA libraries were constructed according to the manufacturer's protocol. RNA-seq was performed by using the Illumina Novaseq platform. The reference genome of SARS-CoV-2 (MN908947) was downloaded from https://www.ncbi.nlm.nih.gov/nuccore /MN908947. After removing low-quality reads, remaining Illumina sequence reads were mapped to human (GRCh38) and SARS2 genome by using HISAT2.1.0 with parameters:—rna-strandness RF–dta. RNA-seq coverage was visualized by using Integrative Genomics Viewer (IGV). To quantify the expression levels of SARS2 genes, RPKM of each virus genes and GFP gene were calculated. Heatmaps were drawn by using R package "pheatmap" (https://www.r-project.org). To quantify the junction-reads from subgenomic RNAs, the STAR2.7.5c was used for reads mapping. The junction-reads was defined and collected as described[45]. A Sankey diagram was drawn by using R packages named "networkD3" and "dplyr".

## IFN-β, neutralizing antibody and drug treatment

To assess the antiviral efficacies of the materials, $1\times10^4$ Caco-2-N cells were seeded into 96-well plates. After 12h, cells were infected with SARS-CoV-2 GFPΔN virus at MOI of 0.05. For neutralizing antibody treatment, virus was incubated with neutralizing antibody for 1 hour at 37˚C before infection. For IFN-β (Sino Bioligical, 10704-HNAS-5) test, cells were pretreated with IFN-β for 8 hours before infection. For remdesivir (MedChemExpress, HY-104077), lopinavir (biochempartner, BCP01395) or ritonavir (biochempartner, BCP03777) treatment, drugs were added simultaneously upon infection. Two days after infection, flow cytometry was performed to analyze GFP positive rate. The 50% inhibitory concentrations (IC50; compound concentration required to inhibit viral replication by 50% reduction of GFP

positive cells) were determined using logarithmic interpolation using GraphPad Prism software version 7.0.

### Cell viability assay

Caco-2-N cells were seeded into 96-well plate ($1 \times 10^3$ cells/well). After 12 hours, cells were treated with drugs with different concentrations. Cell viability was measured two days post treatment with CellTiter-Glo Luminescent Cell Viability Assay kit (Promega, G7570) following standard protocol. In brief, cells in 100 μl culture medium were added with 100 μl CellTiter-Glo reagent. After 15 minutes, luminescence was recorded with GloMax (Promega). $CC_{50}$ was determined using logarithmic interpolation using GraphPad Prism software version 7.0.

### Flow cytometry analysis

Cells were detached in PBS containing 0.02% EDTA and then washed once with cold PBS. Cells were then fixed in 4% PFA for 30 minutes at room temperature. Fixed cells were resuspended in PBS and analyzed by LSRFortessa SORP (BD Biosciences) and FlowJo software.

### Western blotting

Sodium dodecyl sulfate-polyacrylamide gel electrophoresis (SDS-PAGE) immunoblotting was conducted as follows: After trypsinization and cell pelleting at 1500 r/m for 10 min, whole-cell lysates were harvested in cell lysis buffer (50 mM Tris-HCl [pH 7.5], 150mM NaCl, 1% NP-40, 1mM EDTA) supplemented with protease inhibitor cocktail (Sigma). Lysates were electrophoresed in 4–12% polyacrylamide gels and transferred onto PVDF membrane. The blots were blocked at room temperature for 0.5 h using 5% nonfat milk in 1× phosphate-buffered saline (PBS) containing 0.1% (v/v) Tween 20. The blots were exposed to primary antibodies anti-N (05–0154, AbMax), S (40589-T62, Sino Biological), β-Tubulin (CW0098, CWBIO), Flag (F7425, Sigma), ACE2 (10108-T24, Sino Biological) in 5% nonfat milk in 1×PBS containing 0.1% Tween 20 for 2 h. The blots were then washed in 1×PBS containing 0.1% Tween 20. After 1h exposure to HRP-conjugated secondary antibodies and subsequent washes were performed as described for the primary antibodies. Membranes were visualized using the Luminescent image analyzer (GE).

### Antiviral screening

Twelve hours prior to infection for the antiviral screening $5 \times 10^4$ Caco-2-N$^{int}$ cells were seeded in 96 well plates. The next day, a single dilution of each compound of the Topscience Natural Product Library at 5 μM final concentration was added to the cells (50 μL/well). DMSO or remdesivir (3.5μM) were included in each plate as the internal control. After 2 hours, 50 μL of virus was added to the wells at MOI 0.05. Two days after infection, cells were collected for flow cytometry analysis to determine the GFP expression.

### Evaluation of antiviral activity using authentic SARS-CoV-2 virus

A549 cells stably expressing human ACE2 were seeded in a 96-well plate ($4 \times 10^4$ cells/well). Next day, cells were treated with drugs (Lycorine chloride (TargetMol, T2774), Tubeimoside I (TargetMol, T2715), Nigericin sodium (TargetMol, T3092), Monensin sodium (MedChemExpress, HY-N0150), Salinomycin (MedChemExpress, HY-15597)) of different concentration for 2 hours prior to infection. Cells were infected with SARS-CoV-2 at an MOI of 1 for 1 h, washed three times with PBS, and incubated in 2% FBS culture medium for 24 h for viral antigen staining. Cells were fixed with 4% paraformaldehyde in PBS, permeablized with 0.2%

Triton X-100, and incubated with the rabbit polyclonal antibody against SARS-CoV nucleocapsid protein (Rockland, 200-401-A50, 1 μg/ml) at 4˚C overnight. After three washes, cells were incubated with the secondary goat anti-rabbit antibody conjugated with Alexa Fluor 555 (Thermo #A32732, 2 μg/ml) for 2 h at room temperature, followed by staining with 4',6-diamidino-2-phenylindole (DAPI). Images were collected using an Operetta High Content Imaging System (PerkinElmer). For high content imaging, two biological replicates for each concentration of drug were scanned and five representative fields were selected for each well of 96-well plates. Image analysis was performed using the PerkinElmer Harmony high-content analysis software 4.9. Cells were automatically identified by DAPI (nuclei). Mean fluorescent intensity of channel Alexa 555 (viral nucleocapsid) of each cell were subsequently calculated, respectively. For the 0% inhibition control, cells were infected in the presence of vehicle only. The $IC_{50}$ value was defined as the concentration at which there was a 50% decrease in N protein expression. Data were analyzed using GraphPad Prism 7.0. The $IC_{50}$ values were calculated by nonlinear regression analysis using the dose-response (variable slope) equation (four parameter logistic equation).

### SARS-CoV-2 infection and biosafety

The SARS-CoV-2 strain nCoV-SH01 (GenBank accession no. MT121215) was isolated from a COVID-19 patient and propagated in Vero E6 cells for use. The Institutional Biosafety Committee (IBC) of Tsinghua University has approved the SARS-CoV-2 trVLP cell culture system use in BSL-2 if the Caco-2-N$^{int}$ cells were used for infection by SARS-CoV-2 trVLP within 8 passages. All other experiments in this study involving authentic virus or SARS-CoV-2 trVLP infections were performed in the biosafety level 3 facility of Fudan University following all regulations.

### Statistical analysis

Student's $t$ test or one-way analysis of variance (ANOVA) with Tukey's honestly significant difference (HSD) test was used to test for statistical significance of the differences between the different group parameters. $P$ values of less than 0.05 were considered statistically significant.

### Supporting information

**S1 Fig. Generation of Caco-2 cell expressing SARS-CoV-2 N by lentiviral transduction.** (A) Scheme depicting the bicistronic lentiviral constructs for expressing SARS-CoV-2 N protein with C-terminal Flag tag. (B) Representative flow cytometry plots demonstrating efficient lentivirus transduction. Caco-2 cells were transduced with pLVX-N-Flag-IRES-mCherry or not transduced. Flow cytometric analysis was performed 4 d following transduction to quantify the frequencies of N-expressing cells. The flow cytometry result was representative of one of three independent experiments.
(TIF)

**S2 Fig. GFP expression in Caco-2-N cells electroporated with SARS-CoV-2 GFP/ΔN RNA.** (A) GFP expression in Caco-2-N cells electroporated with SARS-CoV-2 GFP/ΔN RNA. Caco-2-N cells were electroporated with 20 μg of SARS-CoV-2 GFP/ΔN RNA. From 21h-96h p.t., GFP expression in the cells was observed with microscopy. (B) GFP expression was quantified by flowcytometry at 96h post transfection of the RNA. This experiment was representative of three independent experiments.
(TIF)

**S3 Fig. Characterization of the genetic stability of SARS-CoV-2 GFP/ΔN virus.** (A) RT-PCR products from P10 virus infected cell passage were cloned into pEASY-Blunt vector, and 12 colonies were randomly chosen for DNA sequences analysis. Multiple deletions were detected in the amplicon. (B) Categories of mapped reads from P1 and P10 virus infected Caco-2-N cells. (C) Canonical discontinuous transcription (top) that is mediated by TRS-L (TRS in the leader) and TRS-B (TRS in the body). Quantification of junction-reads from canonical discontinuous transcripts post P1 and P10 virus infection.
(TIF)

**S4 Fig. Identification of host factors associated with N protein and phosphorylation on N protein by mass spectrometry.** (A) Flag tagged N protein was immunoprecipitated from Caco-2-N cells infected with recombinant SARS-CoV-2 GFP/ΔN trVLP using Flag antibody, and the proteins were analyzed on SDS-PAGE gel. The proteins were visualized by Coomassie blue staining. N, G3BP1 and G3BP2 were labelled. (B) Phosphorylated peptides of N protein derived from Caco-2-N cells. Caco-2-N cells in which N was C-terminal Flag-tagged were collected and cell lysates were immunoprecipitated with anti-Flag coupled beads. Phosphorylated peptides of the immunoprecipitates were analyzed by mass spectrometry.
(TIF)

**S1 Table. Primers used in this study.**
(XLSX)

**S2 Table. Mass spectrometry analysis of host factors associated with N protein.**
(XLSX)

## Acknowledgments

We thank Di Qu, Xia Cai, Zhiping Sun, Wendong Han, and other colleagues at the Biosafety Level 3 Laboratory of Fudan University for help with experiment design and technical assistance. We thank Dr. Guocai Zhong (Shenzhen Bay Laboratory, Shenzhen, China) for generously providing the recombinant ACE2-Fc protein. We thank Prof. Haiteng Deng and Xianbin Meng in Proteinomics Facility at Technology Center for Protein Sciences, Tsinghua University, for protein MS analysis. We thank Drs. Deyin Guo (Sun Yat-sen University, Guangzhou, China) and Kai Wu for suggestions and revision of the manuscript. We are grateful to other members of the Ding lab for critical discussions and comments on the manuscript.

## Author Contributions

**Conceptualization:** Rong Zhang, Qiang Ding.

**Data curation:** Xiaohui Ju, Jingrui Li, Jiaxing Zhang, Sai Li, Rong Zhang.

**Formal analysis:** Xiaohui Ju, Yunkai Zhu, Jingrui Li, Sai Li, Qiang Ding.

**Funding acquisition:** Qiang Ding.

**Investigation:** Xiaohui Ju, Yunkai Zhu, Yuyan Wang, Jiaxing Zhang, Mingli Gong, Wenlin Ren.

**Methodology:** Jingrui Li, Jin Zhong, Qiangfeng Cliff Zhang.

**Project administration:** Xiaohui Ju, Qiang Ding.

**Resources:** Jingrui Li, Linqi Zhang, Rong Zhang.

**Software:** Jingrui Li, Jiaxing Zhang, Sai Li.

**Supervision:** Sai Li, Linqi Zhang, Qiangfeng Cliff Zhang, Rong Zhang, Qiang Ding.

**Validation:** Yunkai Zhu.

**Visualization:** Xiaohui Ju, Jingrui Li, Qiang Ding.

**Writing – original draft:** Xiaohui Ju, Qiang Ding.

**Writing – review & editing:** Jin Zhong, Qiang Ding.

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
