## [Decision Letter · Decision Letter 0]

22 Feb 2021

Dear Dr Ding,

Thank you very much for submitting your manuscript "A novel cell culture system modeling the SARS-CoV-2 life cycle" for consideration at PLOS Pathogens. As with all papers reviewed by the journal, your manuscript was reviewed by members of the editorial board and by several independent reviewers. The reviewers appreciated the attention to an important topic. Based on the reviews, we are likely to accept this manuscript for publication, providing that you modify the manuscript according to the review recommendations.

Both reviewers thought that this biologically contained cell culture model for SARS-CoV0-2 was an important advance that has practical implications for the field. They only listed minor concerns that will help add clarity and improve the readability of the manuscript. Please address these concerns constructively.  Given the importance of this manuscript for the field, I encourage you to return a revised manuscript as soon as possible, so I can accept it editorially.   

Sincerely,

Benhur Lee

Section Editor

PLOS Pathogens

Benhur Lee

Section Editor

PLOS Pathogens

Kasturi Haldar

Editor-in-Chief

PLOS Pathogens

orcid.org/0000-0001-5065-158X

Michael Malim

Editor-in-Chief

PLOS Pathogens

orcid.org/0000-0002-7699-2064

Reviewer Comments (if any, and for reference):

Reviewer's Responses to Questions

**Part I - Summary**

Reviewer #1: In this report, Ju et al., generate and characterise a novel model of SARS-CoV-2 infection, which could be used outside of BSL3 conditions. Through the use of reverse genetics, the team successfully replaced the essential N gene with a fluorescent reporter and stocks generated by infection of cells expression N in trans, complementing the deleted N. To further extend the biosafety of this reagent, Ju et al., exploit the protein splicing intein system to split the in trans N to two genetic fragments, reducing likelihood of recombination and generation of fully infectious SARS-COV-2 outside BSL3 conditions. The team also carry out  extensive passaging and sequencing to identify potential changes yielding infectious virus but none were detected. Finally,  Ju et al., use this safer reporter system to probe the biology of SARS-CoV-2 N and identify novel antiviral compounds against SARS-CoV-2.  This work is an important and timely report on a reagent that would be helpful in our understanding of SARS-CoV-2 infection and identification of potential therapeutic interventions.

Reviewer #2: In this manuscript, Ju et al., developed an N gene-based system to produce a transcription/replication competent SARS-CoV-2 virus-like particles in which the N gene is replaced by the GFP coding sequence. In addition, authors utilized a protein splicing technique to express a functional SARS-CoV-2 N protein through trans-splicing of two independently expressed N subunits. Thus the biosafety of using this VLP is further reduced and can be performed under a BSL-2 condition. The N-complemented VLPs recapitulated several properties of the authentic virus, such as the transcriptome profiling, receptor binding and neutralization ability, and drug sensitivity to known antivirals of SARS-CoV-2. Moreover, several novel antivirals were discovered through the high-throughput screening using the N-complemented VLPs. Overall, the system is well designed and provides a convening way to screen potential antivirals of SARS-CoV-2 in a BSL-2 laboratory.

**Part II – Major Issues: Key Experiments Required for Acceptance**

Reviewer #1: N/A

Reviewer #2: The manuscript is well organized and phrased, and is recommended to be published if several points listed below can be addressed.

1. Figure 1A: The scheme of the viral genome is incorrect. The 3' end contains poly(A) rather than poly(T).

2. line 187-188: Although ORF6, ORF7, and ORF8 genes were lost after 10 passages, the importance of these accessary proteins in viral infection is still unknown. Particularly, the experiments were conducted in a highly permissive cell line. It's better to claim that these proteins might be dispensable for SARS-CoV-2 after its adaptation into certain cell lines.

3. line 228-230: The trans-splicing design is smart and the result is intriguingly. However the introduced G212C mutation is close to R203K/G204R substitutions carried by the recent strains. The possible impact of G212C on the N protein bearing R203K/G204R should be considered and discussed.

4. line 53: Several vaccines are available by Feb 2021. The statement is better to be updated.

5. line 69: ....response and....

6. line 235: structural proteins "and" the primary....

**Part III – Minor Issues: Editorial and Data Presentation Modifications**

Reviewer #1: I have a few minor concerns surrounding presentation and discussion. The authors could pay particular attention to questions surrounding biosafety of the work carried out and any proposed changes to biosafety for SARS-CoV-2 work using this BSL2-compatible system (points 10 and 13).

Abstract:

1. update first line (and intro) regarding no antivirals and treatments when a number of therapies (e/g dexamethasone) and vaccines (e.g. Moderna mRNA) have been licensed across the world.

2. Should be BSL-3 (check for minor errors, BLS3)

3. Is the reporter is GFP or  enhanced GFP? This should be clarified in the methods

Results:

4. How did you divide the genome up into cDNA fragments? What logic to decide where the junctions are?

5. 'T7 promotor' should read T7 DNA-dep RNA polymerase promotor (at least once during definition)

6. What is the rationale for using Caco-2 cells over Vero derivatives?

7. Can the authors speculate on the reasoning why the GFP gene was lost?

8. FLAG tag should be capitalised throughout (or at least keep it one style or the other for consistency)

Discussion:

9. The discussion around proteomic interaction partners would be helped by reference to the earlier literature on other coronaviruses, e.g. https://www.ncbi.nlm.nih.gov/pmc/articles/PMC3754094/ Emmott et al., 2013 (which include description of N/G3BP binding)

10. Additionally, the discussion would be helped (and the biosafety implications furthered) with greater conversation surrounding the potential for the reporter to recombine with infectious virus in BSL2 settings. The reviewers that this is likely very rare but is of potential importance if used widely. Have the authors attempting co-infection and recombination analysis with cells expressing the reporter virus with WT SARS-CoV-2 or other related beta coronaviruses? If the authors do not think this an issue they should say in the discussion.

Legends/methods:

11. Why was ACE2 not observed in Vero cell lysates by your western blot? Is this antibody specific to human? Or is it simply an expression level issue?

12. Were the transduced lines selected using an antibiotic like puromycin? I don't see mention of that here but perhaps none was carried out.

13. Can the authors update the methods describing the level Biosafety of each kind of experiments were performed at. I.e at what stage if any did they move out of the BSL3 (if at all)?

14. Do the authors plan on depositing these reagents in an open repository or how best to request such valuable research reagents? It would be nice if this were addressed.

Reviewer #2: (No Response)

PLOS authors have the option to publish the peer review history of their article (what does this mean?). If published, this will include your full peer review and any attached files.

Reviewer #1: No

Reviewer #2: No

Figure Files:

Data Requirements:

Reproducibility:

References:

---

## [Editor Report · Decision Letter 1]

1 Mar 2021

Dear Dr Ding,

We are pleased to inform you that your manuscript 'A novel cell culture system modeling the SARS-CoV-2 life cycle' has been provisionally accepted for publication in PLOS Pathogens.

Best regards,

Benhur Lee

Section Editor

PLOS Pathogens

Benhur Lee

Section Editor

PLOS Pathogens

Kasturi Haldar

Editor-in-Chief

PLOS Pathogens

orcid.org/0000-0001-5065-158X

Michael Malim

Editor-in-Chief

PLOS Pathogens

orcid.org/0000-0002-7699-2064
---

## [Editor Report · Acceptance letter]

10 Mar 2021

Dear Dr Ding,

We are delighted to inform you that your manuscript, "A novel cell culture system modeling the SARS-CoV-2 life cycle," has been formally accepted for publication in PLOS Pathogens.

Best regards,

Kasturi Haldar

Editor-in-Chief

PLOS Pathogens

orcid.org/0000-0001-5065-158X

Michael Malim

Editor-in-Chief

PLOS Pathogens

orcid.org/0000-0002-7699-2064